# Adolescents with a Favorable Mediterranean-Style-Based Pattern Show Higher Cognitive and Academic Achievement: A Cluster Analysis—The Cogni-Action Project

**DOI:** 10.3390/nu16050608

**Published:** 2024-02-23

**Authors:** Humberto Peña-Jorquera, Ricardo Martínez-Flores, Juan Pablo Espinoza-Puelles, José Francisco López-Gil, Gerson Ferrari, Rafael Zapata-Lamana, Mara Cristina Lofrano-Prado, Leslie Landaeta-Díaz, Igor Cigarroa, Samuel Durán-Agüero, Carlos Cristi-Montero

**Affiliations:** 1IRyS Group, Physical Education School, Pontificia Universidad Católica de Valparaíso, Valparaíso 2530388, Chile; humberto.apj@gmail.com (H.P.-J.); ricardo.antonio.martinezf@gmail.com (R.M.-F.); juan.espinoza.p@mail.pucv.cl (J.P.E.-P.); 2One Health Research Group, Universidad de Las Américas, Quito 170517, Ecuador; josefranciscolopezgil@gmail.com; 3Faculty of Health Sciences, Universidad Autónoma de Chile, Av. Pedro de Valdivia 425, Providencia 7500912, Chile; gerson.demoraes@usach.cl; 4School of Physical Activity, Sports and Health Sciences, Universidad de Santiago de Chile (USACH), Santiago 9170022, Chile; 5School of Education, Universidad de Concepción, Los Ángeles 4440000, Chile; rafaelzapata@udec.cl; 6Departments of Kinesiology and Psychology, California State University, San Bernardino, CA 92407, USA; mara.lofrano@csusb.edu; 7Faculty of Health and Social Sciences, Universidad de las Américas, Santiago 7500975, Chile; llandaeta@udla.cl; 8Core in Environmental and Food Sciences, Universidad de las Américas, Santiago 7500975, Chile; 9School of Kinesiology, Faculty of Health, Universidad Santo Tomás, Los Ángeles 4440000, Chile; icigarroa@santotomas.cl; 10Faculty of Health Sciences, Universidad Arturo Prat, Victoria 4720000, Chile; 11School of Nutrition and Dietetics, Faculty of Health Care Sciences, Universidad San Sebastián, Santiago 8330106, Chile; samuel.duran@uss.cl

**Keywords:** nutrition, learning, cognition, brain function, youth, school, lifestyle

## Abstract

A Mediterranean diet (MedDiet) has emerged as a crucial dietary choice, not only in attenuating various adolescents’ metabolic health issues but it has also been associated with improved cognitive and academic achievement. However, few studies have established patterns of food consumption linked to both cognitive and academic achievement in adolescents living in a developing country with non-Mediterranean-based food. **Methods:** This cross-sectional study included 1296 Chilean adolescents (50% girls) aged 10–14 years. The MedDiet Quality Index was used to assess adherence to the MedDiet in children and adolescents. Through cluster analysis, four distinct dietary patterns were identified: Western diet (WD = 4.3%); low fruit and vegetables, high-sugar diet (LFV-HSD = 28.2%); low fruit and vegetables, low-sugar diet (LFV-LSD = 42.2%); and the MedDiet (25.3%). A mixed-model analysis was conducted to compare these clusters and their relationships with cognitive and academic achievements. Principal component analysis was performed to identify four primary cognitive domains: working memory, cognitive flexibility, inhibitory control, and fluid reasoning. Academic achievement was determined with five school subjects (Language, English, Mathematics, Science, and History) and included the Academic-PISA score derived from the mean scores in Language, Mathematics, and Science. **Results:** A marked difference was observed between the four clusters, which was mainly related to the consumption of sugar, ultra-processed foods, fruits, and vegetables. According to cognitive performance, the MedDiet group showed higher performance across all domains than the LFV-HSD, LFV-LSD, and WD groups. Regarding academic achievement, the WD underperformed in all analyses compared to the other groups, while the MedDiet was the unique profile that achieved a positive difference in all academic subjects compared to the WD and LFV-HSD groups (*p* < 0.05). **Conclusions:** These findings suggest that higher adherence to Mediterranean-style-based patterns and better food quality choices are associated with improved cognitive and academic achievements.

## 1. Introduction

Healthy dietary food patterns have been shown to significantly reduce the risk of various non-communicable chronic diseases and improve the treatment of specific risk factors [1,2]. Different components present in plant sources, the reduction in ultra-processed food consumption, avoiding sugar, including a quality protein intake, and preferring whole-food alternatives, such as fruits and vegetables, could potentially enhance this health setting [3]. Conversely, adopting unhealthy dietary habits and opting for low-quality food have been frequently associated with higher health risks and lower cognitive and academic achievement [4,5].

In this context, the Mediterranean diet (MedDiet) is recognized as a balanced and healthy dietary pattern [6], characterized by the inclusion of extra-virgin olive oil, vegetables, fruits, cereals, nuts, and legumes; a moderate intake of fish, other meat, and dairy products; and a low intake of eggs and sweets [7]. Along with this, recommendations encourage the reduced consumption of processed and sweet foods, alcohol, and tobacco. In contrast, the Western diet (WD) is characterized by the presence of ultra-processed foods, refined grains, and sugar, including high-sugar drinks, candies, and sweets. Both dietary patterns have been well-studied in terms of their relationship with cognitive and academic achievement [8,9].

On the one hand, a WD has been inversely associated with cognitive, academic, and brain development in children, causing increased brain damage in the hippocampus and cerebral prefrontal cortex [10,11]. On the other hand, the MedDiet has been frequently associated with improvements in different aspects of well-being, while also being a supporting influence on cognition and academic achievement [8,9]. In terms of cognitive performance, the MedDiet is associated with elevated consumption of phospholipids, which are essential for maintaining the electrical potential of neuronal membranes and memory formation [12]. In addition, the consumption of omega-3, a fatty acid highly present in the MedDiet (e.g., oily fish, walnuts), acts as a protective factor in cognitive impairment and increases cerebral blood flow [12,13].

Previous research has consistently shown that cognitive performance has a direct impact on academic achievement [14,15]. In this sense, previous studies indicate that higher adherence to the MedDiet was associated with greater academic achievement in children and adolescents [16,17]. However, the relationship between MedDiet and cognitive and academic achievement in youth has not been exhaustively studied in Latin American countries with a non-Mediterranean food pattern culture. Therefore, the acquisition of such patterns and their benefits in this area for this type of population remains unexplored.

In non-Mediterranean countries, such as most countries worldwide, full MedDiet adherence is difficult not only due to cultural factors but also to family socioeconomic status, physical activity practice, physical fitness levels, parental influence, and acquired eating habits, among others [18,19,20,21]. It is important to note that Chile is one of few countries that has a temperate coastal Mediterranean-type climate, particularly prevalent in our sample zone (−33.02457, −71.55183) [22]. The region features accessible agriculture with foods similar to those of Mediterranean countries, including biodiversity and demographic locations [23,24]. Despite these favorable conditions, there is no Mediterranean food culture. Previous studies indicated that only 10% of the Chilean adult population adheres to this pattern [23]. Regarding Chilean youth, the literature is limited; however, in the Latin American context, a study conducted in Colombia [25] reported a MedDiet adherence rate of 12% in children and adolescents. In this sense, it is important to mention that a strategy to improve adherence to the MedDiet can be exposure to this dietary pattern from childhood and adolescence, considering that it is a sensitive stage in the creation of eating habits that can contribute to improving the health of the population [20].

Consequently, to comprehensively investigate MedDiet adherence and its potential benefits in the cultures of non-Mediterranean-based countries, it is necessary not only to assess full adherence but also to explore intermediate-specific adherence patterns. These intermediate patterns result from various food combinations that conform to different Mediterranean-style eating patterns [26]. An effective approach to achieve this is through latent class analysis, which enables the identification of nuanced patterns of adherence, taking into account the diverse range of foods typically associated with the MedDiet [27]. This information can guide the incorporation of this dietary pattern into the population by analyzing the influences of various food combinations and their relationship to cognitive and academic achievement.

Therefore, considering the lack of evidence in this age group, particularly in the dietary cultures of non-Mediterranean-based countries, this study aimed to explore the association between different Mediterranean-style-based patterns and cognitive and academic achievement in a large sample of Chilean adolescents.

## 2. Methods

This cross-sectional study is part of the Cogni-Action Project. The project was conducted between March 2017 and October 2019 [28]. The project was approved by the Bioethics and Biosafety Committee of the Ethics Committee of the Pontificia Universidad Católica de Valparaíso (BIOEPUCV-H103-2016) and conducted in accordance with the guidelines of the Declaration of Helsinki. The present study was performed according to the STROBE guidelines. Written consent was obtained from the school principal, parents/guardians, and participants prior to participation.

### 2.1. Participants

An open invitation to schools from Valparaíso was extended after consulting the Chilean Ministry of Education’s database. This study included a total of 1296 adolescent boys and girls (1:1 ratio; 50% girls) aged between 10 and 14 years, corresponding to the 5th- to 8th-grade levels, from public, subsidized (i.e., schools receiving government financial support), and private schools in Valparaiso, Chile. Power calculation was based on the total enrolment of children and adolescents (from 5th to 8th grade) indicated by the Chilean Ministry of Education in the year 2016 (universe *n* = 951,962), assuming an alpha error of 5%; confidence interval of 99; 50% of heterogeneity; and with a 20% dropout [28].

### 2.2. Procedures and Measurements

For this project, participants were evaluated at school in two four-hour sessions separated by eight days. Body weight, height, waist circumference, eating habits, sociodemographic information, and a complete cognitive battery were assessed in the first session. The second session evaluated physical fitness through the well-documented ALPHA-fitness test battery [29]. Trained instructors from our research team performed the measurements. The academic achievement variables were obtained from each student’s school.

### 2.3. Mediterranean-Style-Based Pattern

To establish the nutritional quality of the children, we applied the Mediterranean Diet Quality Index (KIDMED) for children and adolescents [30], which allowed us to establish the adherence and frequency of the foods of a MedDiet. In this sense, after a sensitivity analysis explained in the Section 2.7, nine items of the questionnaire were included in the analysis: (1) second serving of fruit daily and (2) fresh or cooked vegetables 1/day; (3) regular fish consumption (at least 2–3/week); (4) 1/week fast-food (hamburger) restaurant; (5) pasta or rice almost daily (≥5/week); (6) regular nut consumption (at least 2–3/week); (7) no breakfast; (8) commercially baked goods or pastries for breakfast; (9) sweets and candy several times a day. To abbreviate the terms, the following items were referred to as “Second fruit” for “second serving of fruit daily”; “Second vegetable” for “fresh or cooked vegetable 1/day”; “Fish” for “regular fish consumption (at least 2–3/week)”; “Junk food” for “1/week fast-food (hamburger) restaurant”; “Pasta” for “pasta or rice almost daily (≥5/week)”; “Nuts” for “regular nut consumption (at least 2–3/week)”; “Skip breakfast” for “no breakfast”; “Pastry breakfast” for “commercially baked goods or pastries for breakfast”; “Sweet” for “sweets and candy several times a day”.

### 2.4. Academic Achievement (ACA)

According to school records, ACA was established through five school subjects (Language, English, Mathematics, Science, and History), and final school grades were obtained from official records. In Chile, the grade scoring range is between 1 and 7 points, and Language, Mathematics, and Science are the main subjects included in the Programme for International Student Assessment (PISA) by the Organization for Economic Cooperation and Development (OECD) [31]. Grades are expressed on a national scale, ranging from 1 to 7. The average of these three subjects was also computed (Academic-PISA score).

### 2.5. Cognitive Performance

The NeuroCognitive Performance Test (NCPT) from Lumos Labs, Inc. (San Francisco, CA, USA) [32] assessed cognitive functioning. The NCPT is a tool designed for the assessment of various domains of cognitive performance. It runs on a web-based platform and has demonstrated reliability and validity in measuring cognitive performance among adults, aligning well with traditional pencil-and-paper assessments [32]. It is crucial to acknowledge the gap in validation studies specific to children and adolescents; however, the Grand Index, a compilation of scores normalized across NCPT tasks, demonstrates robust test-retest reliability (r = 0.78) within the age range of 8–15 years (unpublished data) [33].

This test was conducted in groups of 25 students, each with a laptop. This test is a brief, repeatable, web-based platform to measure several cognitive domains: “Trail Making A and B” assessing attention, cognitive flexibility, and processing speed; the “Forward Memory Span” and the “Reverse Memory Span” evaluating short-term visual memory and working memory; the “Go/No-Go” test assessing inhibitory control and processing speed; the “Problem-solving” indicating quantitative and analogical reasoning; the “Digit Symbol Coding” valuing processing speed; and finally, the “Progressive Matrices” assessing problem-solving and reasoning/intelligence [32]. Each test was scaled following a normal inverse transformation of the percentile rank and summed to obtain a global cognitive score [34].

### 2.6. Covariates

Three covariates (sex, maturity, and global fitness score) were included in the analysis owing to their relevance to the outcome. Sex and maturation are relevant factors in cognitive and brain development [35]. Maturity was calculated according to the peak height velocity (PHV) by subtracting the PHV age from chronological age [36]. A global fitness score was included because of the positive association between fitness and cognitive performance [34]. This index was calculated from the cardiorespiratory fitness measured using the 20 m shuttle run test. Muscle strength of the upper and lower limbs was measured using a handgrip and standing long jump, respectively. Speed agility was assessed using the 4 × 10 m shuttle run test with the ALPHA-fitness test battery [29]. More details on the Cogni-Action Project measures have been published elsewhere [28].

### 2.7. Statistical Analysis

Firstly, the dataset was subjected to imputation based on the non-parametric missing value method using random forest through the “missForest” R package [37]. This imputation algorithm is suitable for handling mixed data types (numerical or categorical variables). It stands out for not requiring preprocessing steps or making assumptions of a parametric nature while delivering a strong predictive capability. After imputation, the analysis considered the total sample size (*n* = 1296). The average percentage of missing data across the variables used in this study was low, for instance, the second serving of fruit daily (11.7%), fresh or cooked vegetables 1/day (12.5%), regular fish consumption (at least 2–3/week) (13.3%), 1/week fast-food (hamburger) restaurant (13.0%), pasta or rice almost daily (≥5/week) (11.8%), regular nut consumption (at least 2–3/week) (12.6%), no breakfast (13.7%), commercially baked goods or pastries for breakfast (13.5%), sweets and candy several times a day (12.7%), global fitness score (24.5%), Language score (1.5%), English score (2.2%), Science score (1.7%), History score (1.6%), and PHV (1.2%); the estimation error was 0.01% for numeric variables and 0.02% for factors.

Secondly, a principal component analysis was conducted to identify and establish four cognitive domains: working memory (WM), cognitive flexibility (CF), inhibitory control (IC), and fluid reasoning (FR). These factors were determined based on the specific characteristics of our cognitive tasks and the existing literature, considering the sample size of our previous publications [34]. Thus, four fixed components were established using orthogonal varimax rotation, explaining 71.2% of the variance (WM = 23.3%, CF = 18.1%, IC = 16.8%, and FR = 13.0%). The assumption check for Bartlett’s test of sphericity was significant (*p* < 0.001). Analyses were performed using Jamovi based on the “psych” Package for R [38].

Thirdly, a correlation matrix analysis was performed to determine which items of the KIDMED questionnaire were related to cognitive domains and academic achievement, and 9 of the 16 items were included in our analysis (those mentioned in the “Dietary patterns” section [39]). It is important to note that the items “Fruit or fruit juice daily” and “Fresh or cooked vegetables daily” were omitted, as we included “Second serving of fruit daily” and “Fresh or cooked vegetables 1/day”. The results of the correlation matrix can be found in the Appendix A.

Fourthly, a latent class analysis (LCA) was performed utilizing the nine items in our model to determine eating patterns, and four clusters were established. Generally, LCA is a statistical procedure that facilitates the identification of distinct groups, enabling the discovery of unobserved heterogeneity within a given dataset [27]. To decide the number of clusters indicated, a sensitivity analysis was performed with an elbow plot, which was employed to contrast various indicators of model fit (i.e., Akaike Information Criterion [AIC], Bayesian Information Criterion [BIC], Bozdogan’s Adjusted BIC [ABIC], and Consistent AIC [CAIC]). Analyses were performed using Jamovi based on the ”snowRMM” Package for R [40].

Descriptive statistics are presented as means, standard deviations, frequencies, and percentages (Table 1). Parametric tests (Student’s *t*-test, chi-square test, and mixed models) were used to conduct analyses, as indicated by the central limit theorem for sample sizes of over 500 participants [41]. In parallel, normality distribution was checked visually with a Q–Q plot (quantile–quantile plot) and the Shapiro–Wilk test.

Mixed model analyses were performed to establish the differences between clusters, cognitive domains, and academic scores. To compare the likelihood of a model with the effect included vs. a model with the effect excluded, the likelihood-ratio test (LRT) for the random effect was estimated. A significant value indicates that the model with the random effect is significantly better (in terms of likelihood) than the model without the random effect, and the interclass correlation coefficient (ICC) was estimated. If the value was not significant, the assumptions were checked using ANCOVAs; more information is provided in the Appendix A. School type (k = 3; public, subsidized, and private schools) was used as the random effect. We use this variable because, in Chile, family socioeconomic status is highly predictive of the school type their children attend; thus, low-, middle-, and high-socioeconomic status families send their children to public, subsidized, and private schools, respectively [42]. Post hoc tests were performed using the Bonferroni correction for multiple comparisons. Statistical significance was set at *p* < 0.05. All models were adjusted for the multiple covariates mentioned previously. Mixed models were generated using the statistical software Jamovi version 2.3.18 [43].

## 3. Results

### 3.1. Characteristics of the Participants

A total of 1296 adolescents (50.0% girls) aged 10–14 participated in this study. Based on our analysis, we identified four distinct clusters: WD (4.3%), LFV-HSD (28.2%), LFV-LSD (42.2%), and MED-DIET (25.3%). The frequency detail of each dietary item assessed can be found in the Appendix A. Table 1 shows a complete description of adolescent characteristics. Table 2 details the dietary components evaluated, according to the KIDMED questionnaire, while Table 3 reports the assessment of cognitive performance tasks and academic achievement results. For each table, the information of every cluster was reported.

Figure 1 illustrates the radar plot analysis of the outcomes attributed to each cluster to the specified indicator. Given that the KIDMED indicators represent a dichotomous result, our emphasis was on the positive fulfillment of each of these indicators.

### 3.2. Differences in Cognitive Performance concerning Clusters

Figure 2 shows the significant differences between the four clusters according to the cognitive domains: working memory (LRT = 10.2; *p* = 0.001; ICC = 0.0168), cognitive flexibility (LRT = 8.48; *p* = 0.004; ICC = 0.0173), inhibitory control (LRT = 36.0; *p* = < 0.001; ICC = 0.0610), fluid reasoning (LRT = 3.73, *p* = 0.054, ICC = 0.00995), and total cognitive performance (LRT = 56.4, *p* = < 0.001, ICC = 0.0864). The WD group showed lower performance across all cognitive domains than the other groups. In particular, the WD group reported a negative difference in all domains compared with the MedDiet group (*p* < 0.05) except for inhibitory control (*p* = 0.142). When comparing the MedDiet with the LFV-HSD cluster, the former reported a positive difference in total cognitive performance (*p* < 0.001), cognitive flexibility (*p* = 0.004), inhibitory control (*p* = 0.037), and working memory (*p* < 0.001) but not in fluid reasoning (*p* = 0.705). Lastly, the comparison between the MedDiet and LFV-LSD groups showed no significant differences in any cognitive domain. Details of the post hoc results can be found in the Appendix A.

### 3.3. Differences in Academic Achievement concerning Clusters

Figure 3 shows significant differences between the four clusters in academic achievement: Language (LRT = 0.774; *p* = 0.379; ICC = 0.00252), English (LRT = 71.9; *p* = < 0.001; ICC = 0.0810), Mathematics (LRT = 0.00395; *p* = 0.950; ICC = 0.000190), Science (LRT = 11.7; *p* = < 0.001; ICC = 0.0192), History (LRT = 3.06, *p* = 0.080, ICC = 0.00740), and PISA (LRT = 0.0385; *p* = 0.844; ICC = 0.000577). The WD group underperformed in all academic analyses compared with the other groups. Notably, the MedDiet cluster was the only one that achieved a statistically significant difference (*p* < 0.05) in all school subjects compared to both the WD and LFV-HSD groups. Finally, the comparison between the MedDiet and LFV-LSD groups indicated that the former performed better in all academic analyses than the LFV-LSD group, but the differences were not statistically significant. Details of the post hoc results can be found in the Appendix A.

## 4. Discussion

This study aimed to investigate the relationship between distinct Mediterranean-style-based dietary patterns and cognitive and academic achievement in a large sample of Chilean schoolchildren. Our primary analysis identified four different food patterns: WD, LFV-HSD, LFV-LSD, and MedDiet, based on whether adolescents adhered to or did not adhere to several components related to the MedDiet. These four clusters exhibited key differences in the consumption of fruits and vegetables, breakfast omission, sugar or junk food intake, and fish consumption.

On the one hand, our findings suggest that adhering to a full Mediterranean-style pattern is associated with higher cognitive and academic achievement in comparison with the WD. In line with our findings, several studies have indicated that higher adherence to the MedDiet is associated with higher cognitive performance [8,9,44]. This favorable relationship between a MedDiet and cognitive function is attributed to the nutrient-rich composition of the MedDiet, which includes minerals, fiber, potassium, flavonoids, carotenoids, fatty acids, and polyphenols [45,46,47,48]. In this sense, at the brain level, the MedDiet has been associated with gains in white matter structural connectivity [49], reduced cortical thinness, greater prefrontal cognitive function, increased white matter volume, gray matter, and total brain volume [44,47,50,51].

Regarding our findings on academic performance, the previous literature indicates that greater adherence to the MedDiet has been associated with higher academic achievement [16]. Similarly, a study in a non-Mediterranean country showed the same direction [17]. One possible explanation for this association may be the well-documented relationship between cognitive performance and academic achievement [14,15]. In this case, MedDiet may play a mediating role in the relationship between cognitive performance and academic achievement. Therefore, future studies should explore this phenomenon.

On the other hand, a novelty of this study lies in the exploration of intermediate Mediterranean-style dietary patterns. In general, the two patterns were associated with higher cognitive and academic achievement than WD. Supporting this, the literature shows that a bad quality high-energy density diet, refined carbohydrates, and high saturated fats affect blood flow to the brain [52]. In this sense, our results suggest that the adoption of intermediate patterns based mainly on low sugar consumption could serve as a kick-start in transitioning toward a Mediterranean-style dietary pattern in non-Mediterranean regions.

In particular, both intermediate clusters had a common factor of low fruit and vegetable consumption, which is not surprising in the Chilean context. The Chilean population (90%) does not reach the fiber recommendation and positions bread as the main fiber supplier food with 4.39 ± 3.05 g/day, while fruits and vegetables contribute only 1.54 ± 1.49 and 1.85 ± 1.59 g/day, respectively [53]. However, although both clusters had low fiber intake, one was characterized by high sugar consumption, while the other was not (LFV-HSD vs. LFV-LSD).

In this sense, it is important to note that there were statistically significant differences in cognitive performance (cognitive flexibility *p* = 0.006; working memory *p* = 0.018; inhibitory control *p* = 0.004; total cognitive performance *p* ≤ 0.001) and academic achievement (English *p* = 0.003; History *p* = 0.004; Language *p* = 0.030) in favor of the LFV-LSD cluster (low sugar consumption). Previous studies support these results, as they point to an association between high sugar consumption and poorer cognitive and academic achievement, mainly due to refined carbohydrates, as they negatively affect the frontal, limbic, and hippocampal system functions affecting learning, memory, and cognition [54].

The mechanism underlying this association may be explained by low-grade systemic inflammation caused by high sugar intake and blood sugar levels [55]. In this regard, previous scientific evidence indicates that high blood sugar levels can generate significant brain damage due to brain glucose dysregulation, which can cause various degenerative processes, neuronal cell death, or loss [56]. It is also noted that high consumption of refined sugars is associated with reduced perfusion in the prefrontal cortex and increased perfusion in the hypothalamus and striatum [57]. In this sense, due to the association between brain glucose dysregulation and brain damage, current research calls Alzheimer’s disease “Diabetes type 3” [56].

Based on the results of our research, it is fundamental to recognize that a full Mediterranean-style pattern is difficult to apply in cultures of non-Mediterranean-food countries due to various factors, despite the possible dietary accessibility to this pattern; therefore, it is necessary to explore intermediate Mediterranean-style-based patterns that could also have the potential to be associated with cognitive performance and academic achievement. These intermediate Mediterranean-style-based patterns may be a basic nutritional and public health strategy for obtaining the underlying brain health and educational benefits of the MedDiet. Future research should explore the possible mediating role of intermediate Mediterranean-style-based patterns in the relationship between cognitive functioning and academic achievement to provide a more comprehensive understanding of these complex interconnections.

## 5. Strengths and Limitations

Finally, our study had some limitations. For instance, the cross-sectional nature of our approach precludes the establishment of causality even with meticulous adjustments for pertinent variables. Although we studied dietary patterns, we were unable to investigate the conformation of these patterns (i.e., frequency and time of consumption). A more extensive evaluation over a greater number of days would have provided increased precision for our overall analysis. For this criterion, in future research, we recommend considering this gap in how the precision of food frequency is measured. Thus, a more accurate evaluation of the food frequency and the relationship with adherence to the Mediterranean diet would be obtained. Nonetheless, our study has some strengths, such as its robust analysis methodology for comprehensively understanding the influence of dietary patterns on cognitive and academic achievement. In addition, our analysis comes from a sample of a less-studied age group, contributing to filling gaps in this population. The last is relevant considering that most studies examining this association have been conducted in Mediterranean countries. Additionally, a pivotal strength lies in the incorporation of body mass index, socioeconomic factors, and general fitness as covariates, thereby enhancing the understanding of our analytical framework. Moreover, our study included both adolescents’ cognitive and academic assessments in the same analysis, while most existing evidence only considers one of them at a time.

## 6. Conclusions

This study underscores the potential benefits of adopting both complete and intermediate adherence to MedDiet patterns in a non-Mediterranean adolescent sample, revealing a positive association with cognitive and academic achievement. Specifically, an intermediate Mediterranean-style-based pattern marked by low sugar consumption demonstrated a favorable association compared to WD. Consequently, this study suggests that a gradual transition toward a Mediterranean-style-based dietary approach could emerge as a plausible initial strategy in nutrition and public health, linked to more desirable outcomes in brain health and educational achievements. The impact on children’s and adolescents’ learning may be impressive if future longitudinal studies shed light on the reported relationships and potential causal pathways over time.

## Figures and Tables

**Figure 1 nutrients-16-00608-f001:**
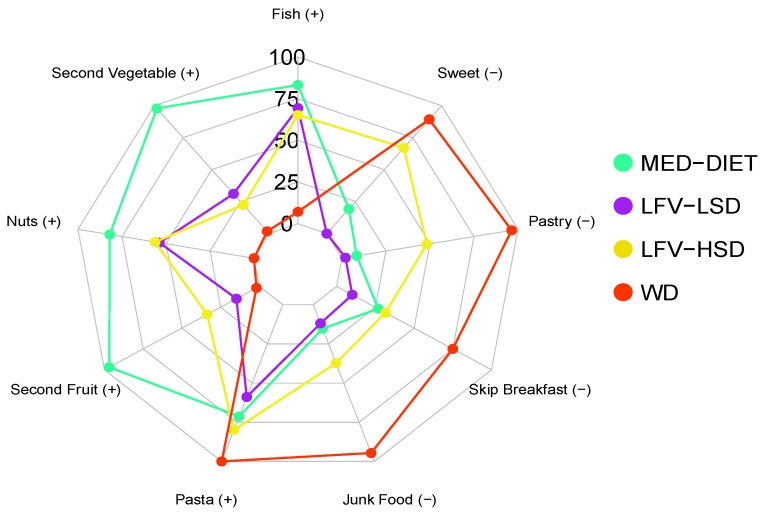
Radar plot of the percentage of adherence to each indicator used in our analysis according to each cluster. MED-DIET: Mediterranean diet; LFV-LSD: Low fruit and vegetables, low-sugar diet; LFV-HSD: Low fruit and vegetables, high-sugar diet; WD: Western diet; (+) Adherence to a healthy indicator; (−) Adherence to an unhealthy indicator.

**Figure 2 nutrients-16-00608-f002:**
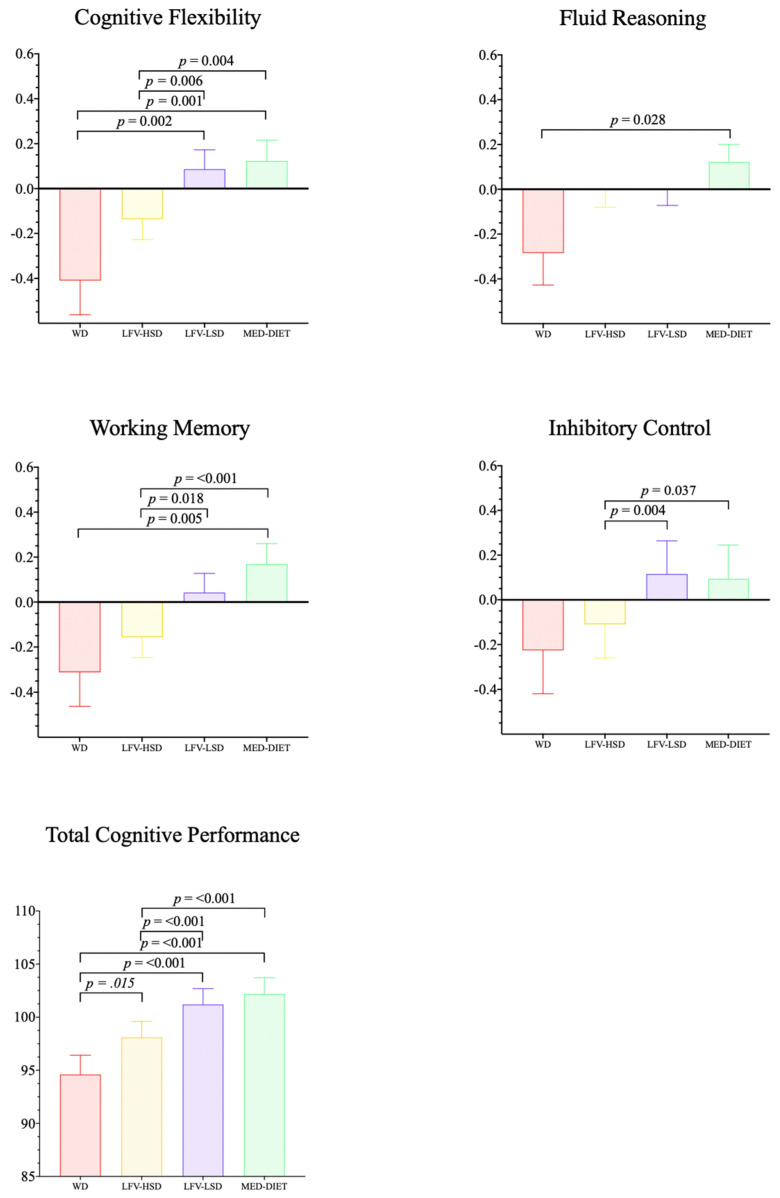
Comparisons between clusters according to cognitive domains. WD = Western diet; LFV–HSD = low fruit and vegetable, high–sugar diet; LFV–LSD = low fruit and vegetable, low–sugar diet; MedDiet = Mediterranean diet.

**Figure 3 nutrients-16-00608-f003:**
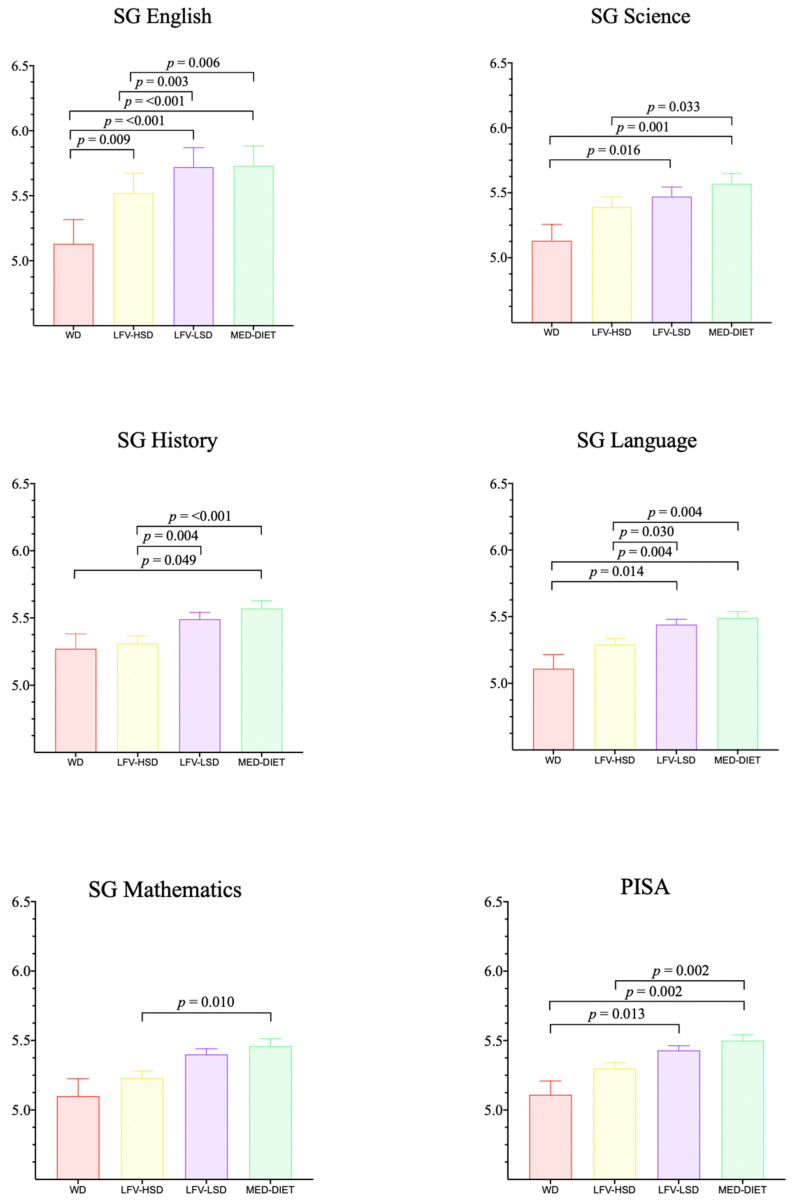
Results of the mixed models for academic achievement. WD, Western diet; LFV–HSD = Low fruit and vegetable, high–sugar diet; LFV–NSD = Low fruit and vegetable, low–sugar diet; MedDiet = Mediterranean diet.

**Table 1 nutrients-16-00608-t001:** Adolescent characteristics (*n* = 1296).

Variables	All (*n* = 1296)	WD (*n* = 56)	LFV-HSD (*n* = 365)	LFV-LSD (*n* = 547)	MED-DIET (*n* = 328)
Sex (boys/girls)	648/648	34/22	166/199	283/264	165/163
Age (years)	11.9 ± 1.2	12.2 ± 1.1	12.3 ± 1.2	12.3 ± 1.1	12.6 ± 1.1
Peak high velocity	−0.41 ± 1.3	−0.79 ± 1.4	−0.39 ± 1.2	−0.51 ± 1.2	−0.21 ± 1.3
Weight	50.9 ± 12.0	50.8 ± 14.9	50.7 ± 12.5	50.6 ± 11.0	51.6 ± 12.4
Height	153 ± 9.31	150 ± 10.2	153 ± 9.4	153 ± 9.1	154 ± 9.3
BMIz	1.02 ± 1.06	1.22 ± 1.1	0.99 ± 1.1	1.1 ± 1.1	0.96 ± 1.1
**School type**					
Public	456 (35.2%)	37 (66.1%)	161 (44.1%)	145 (26.5%)	113 (34.5%)
Subsidized	514 (39.7%)	15 (26.8%)	140 (38.4%)	249 (45.5%)	110 (33.5%)
Private	326 (25.2%)	4 (7.1%)	64 (17.5%)	153 (28.0%)	105 (32.0%)

**Table 2 nutrients-16-00608-t002:** Dietary components of MedDiet (*n* = 1296).

Variables	All (*n* = 1296)	WD (*n* = 56)	LFV-HSD (*n* = 365)	LFV-LSD (*n* = 547)	MED-DIET (*n* = 328)
Second serving of fruit daily					
Adheres	523 (40.4%)	1 (1.8%)	123 (33.7%)	81 (14.8%)	318 (96.9%)
Does not adhere	773 (59.6%)	55 (98.2%)	242 (66.3%)	466 (85.2%)	10 (3.1%)
Fresh or cooked vegetables 1/day					
Adheres	574 (44.3%)	1 (1.8%)	82 (22.5%)	170 (31.1%)	321 (97.9%)
Does not adhere	722 (55.7%)	55 (98.2%)	283 (77.5%)	377 (68.9%)	7 (2.1%)
Regular fish consumption (at least 2–3/week)					
Adheres	894 (69.0%)	4 (7.1%)	238 (65.2%)	379 (69.3%)	273 (83.2%)
Does not adhere	402 (31.0%)	52 (92.9%)	127 (34.8%)	168 (30.7%)	55 (16.8%)
Once/week fast-food (hamburger) restaurant					
Adheres	304 (23.5%)	53 (94.7%)	136 (37.3%)	65 (11.9%)	50 (15.2%)
Does not adhere	992 (76.5%)	3 (5.3%)	229 (62.7%)	482 (88.1%)	278 (84.8%)
Pasta or rice almost daily (≥5/week)					
Adheres	904 (69.8%)	56 (100.0%)	292 (80.0%)	322 (58.9%)	234 (71.3%)
Does not adhere	392 (30.2%)	0 (0.0%)	73 (20.0%)	225 (41.1%)	94 (28.7%)
Regular nut consumption (at least 2–3/week)					
Adheres	768 (59.3%)	0 (0.0%)	205 (56.1%)	294 (53.7%)	269 (82.0%)
Does not adhere	528 (40.7%)	56 (100.0%)	160 (43.9%)	253 (46.3%)	59 (18.0%)
No breakfast					
Adheres	300 (23.1%)	42 (75.0%)	115 (31.5%)	55 (10.1%)	88 (26.8%)
Does not adhere	996 (76.9%)	14 (25.0%)	250 (68.5%)	492 (89.9%)	240 (73.2%)
Commercially baked goods or pastries for breakfast					
Adheres	304 (23.5%)	54 (96.4%)	176 (48.2%)	11 (2.0%)	27 (8.2%)
Does not adhere	992 (76.5%)	2 (3.6%)	189 (51.8%)	536 (98.0%)	301 (91.8%)
Sweets and candy several times a day					
Adheres	358 (27.6%)	50 (89.3%)	245 (67.1%)	0 (0.0%)	63 (19.2%)
Does not adhere	938 (72.4%)	6 (10.7%)	120 (32.9%)	547 (100.0%)	265 (80.8%)

**Table 3 nutrients-16-00608-t003:** Cognitive and Academic measurement (*n* = 1296).

Variables	All (*n* = 1296)	WD (*n* = 56)	LFV-HSD (*n* = 365)	LFV-LSD (*n* = 547)	MED-DIET (*n* = 328)
Cognitive tasks					
Cognitive Flexibility					
Trail-making test A (p)	100.0 ± 14.7	91.4 ± 13.5	97.7 ± 14.5	101.1 ± 14.3	102.2 ± 14.9
Trail-making test B (p)	100.0 ± 14.7	93.3 ± 15.3	97.1 ± 14.1	101.5 ± 14.6	101.8 ± 14.7
Digit coding symbol (p)	100.0 ± 14.7	92.3 ± 13.4	97.2 ± 14.6	101.6 ± 14.5	101.8 ± 14.4
Working Memory					
Memory forward (p)	100.0 ± 14.4	92.0 ± 14.9	97.2 ± 13.6	101.4 ± 14.2	102.1 ± 14.4
Memory reverse (p)	100.0 ± 14.4	94.8 ± 14.6	96.7 ± 14.3	101.2 ± 13.9	102.4 ± 14.2
Inhibitory Control					
Go/No-Go (p)	100.0 ± 14.7	93.9 ± 15.4	98.4 ± 15.5	100.5 ± 14.6	101.9 ± 13.4
Fluid Reasoning					
Problem-solving (p)	100.0 ± 14.5	92.8 ± 13.4	97.4 ± 13.9	101.3 ± 14.3	102.1 ± 14.8
Progressive matrices (p)	100.0 ± 14.3	94.1 ± 11.4	97.6 ± 13.4	101.6 ± 14.6	101.3 ± 14.5
Academic achievement					
English (s)	5.62 ± 0.9	5.16 ± 0.9	5.52 ± 0.9	5.68 ± 0.8	5.72 ± 0.9
History (s)	5.45 ± 0.8	5.20 ± 0.6	5.31 ± 0.8	5.50 ± 0.8	5.56 ± 0.8
Language (s)	5.40 ± 0.8	5.09 ± 0.7	5.31 ± 0.8	5.45 ± 0.8	5.47 ± 0.8
Mathematics (s)	5.35 ± 1.0	5.09 ± 0.9	5.24 ± 1.0	5.42 ± 0.9	5.42 ± 1.0
Science (s)	5.45 ± 0.8	5.10 ± 0.8	5.39 ± 0.9	5.46 ± 0.8	5.54 ± 0.8
Academic-PISA average	5.40 ± 0.8	5.09 ± 0.7	5.31 ± 0.8	5.44 ± 0.7	5.47 ± 0.8

Data are expressed as mean ± standard deviation or count (percentage). The significant *p*-values are shown in bold. (y): years; (p): percentile; (s): score; BMIz: Body Mass Index z-score. Academic achievement Chilean scale up to 7.0 score; Academic-PISA average: Language, Mathematics, and Science average. MedDiet: Mediterranean diet. Student’s *t*-test for independent samples.

## Data Availability

The data presented in this study are available on request from the corresponding author. The data are not publicly available due to ethical concerns (carlos.cristi.montero@gmail.com).

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
