# Peer review of "Adolescents with a Favorable Mediterranean-Style-Based Pattern Show Higher Cognitive and Academic Achievement: A Cluster Analysis—The Cogni-Action Project"

_nutrients, 2024, doi:10.3390/nu16050608_

Round 1

Reviewer 1 Report

Comments and Suggestions for Authors

The paper is very well developed. The topic is original, and the value is enhanced by following the STROBE recommendations and the additional material in the supplement. I missed the comparison of clusters, but that is what is in the supplement.  I feel that the paper is overloaded with advanced statistics but lacks basic information.

More detailed comments are below.

The aim of the study was stated very generally. I suggest adding the main research questions or research problems. Table 1 shows the analysis by gender, which is not discussed further and does not come from the purpose of the study.

I suggest dividing Table 1 into three parts, especially as there is mostly no limitation on the number of tables in MDPI journals. Please add N in the titles of these tables.

A latent class method is given as a clustering method. Please explain in 1-2 sentences the idea of this method.  It is not clear which software was used for each analysis.

Please check the reference to all tables also those provided in the supplement. There is only an overly general comment about additional explanations in the supplement.

The basic information that should be at the beginning of the results about the distribution of clusters in the population was missing. This should also be in the abstract. The main topic is MD and I don't know how often it is selected by adolescents.

I suggest distinguishing thematic sections (subchapters) in the description of the results. Ideally, these would correspond with the suggested research problems, which are lacking at the moment.

In describing the limitations of the study, it is worth emphasising more strongly that the classification of the diet would be better if at least for 1-3 days the products consumed were analysed.  Nevertheless, the aim of the study is to determine the energy of the diet and its composition.

In the title of Figure 1, abbreviations are not explained.

The quality of Figures 2 and 3 should be better.

Reviewer 2 Report

Comments and Suggestions for Authors

This is a complete, well-written manuscript, with clear objectives and scientific solidity. My only concern is the control of extraneous variables in the formation of the groups, since being completely self-reported there is no solid evidence of frequency and times of consumption that can affect adherence to a Mediterranean diet. It is important to note that there were no differences in the mathematics subject. What explanation can be given for this result? I consider that the article needs to answer these couple of issues.
